# On Separability of Loss Functions, and Revisiting Discriminative Vs Generative Models

**Adarsh Prasad**
Machine Learning Dept.
CMU
adarshp@andrew.cmu.edu

**Alexandru Niculescu-Mizil**
NEC Laboratories America
Princeton, NJ, USA
alex@nec-labs.com

**Pradeep Ravikumar**
Machine Learning Dept.
CMU
pradeepr@cs.cmu.edu

## Abstract

We revisit the classical analysis of generative vs discriminative models for general exponential families, and high-dimensional settings. Towards this, we develop novel technical machinery, including a notion of separability of general loss functions, which allow us to provide a general framework to obtain $\ell_\infty$ convergence rates for general $M$-estimators. We use this machinery to analyze $\ell_\infty$ and $\ell_2$ convergence rates of generative and discriminative models, and provide insights into their nuanced behaviors in high-dimensions. Our results are also applicable to differential parameter estimation, where the quantity of interest is the difference between generative model parameters.

## 1   Introduction

Consider the classical conditional generative model setting, where we have a binary random response $Y \in \{0, 1\}$, and a random covariate vector $X \in \mathbb{R}^p$, such that $X|(Y = i) \sim P_{\theta_i}$ for $i \in \{0, 1\}$. Assuming that we know $P(Y)$ and $\{P_{\theta_i}\}_{i=0}^1$, we can use the *Bayes rule* to predict the response $Y$ given covariates $X$. This is said to be the *generative model approach* to classification. Alternatively, consider the conditional distribution $P(Y|X)$ as specified by the Bayes rule, also called the discriminative model corresponding to the generative model specified above. Learning this conditional model directly is said to be the *discriminative model approach* to classification. In a classical paper [8], the authors provided theoretical justification for the common wisdom regarding generative and discriminative models: when the generative model assumptions hold, the generative model estimators initially converge faster as a function of the number of samples, but have the same asymptotic error rate as discriminative models. And when the generative model assumptions do not hold, the discriminative model estimators eventually overtake the generative model estimators. Their analysis however was for the specific generative-discriminative model pair of Naive Bayes, and logistic regression models, and moreover, was not under a high-dimensional sampling regime, when the number of samples could even be smaller than the number of parameters. In this paper, we aim to extend their analysis to these more general settings.

Doing so however required some novel technical and conceptual developments. To motivate the machinery we develop, consider why the Naive Bayes model estimator might initially converge faster. The Naive Bayes model makes the conditional independence assumption that $P(X|Y) = \prod_{s=1}^p P(X_s|Y)$, so that the parameters of each of the conditional distributions $P(X_s|Y)$ for $s \in \{1, \ldots, p\}$ could be estimated independently. The corresponding log-likelihood loss function is thus fully "separable" into multiple components. The logistic regression log-likelihood on the other hand is seemingly much less "separable", and in particular, it does not split into multiple components each of which can be estimated independently. In general, we do not expect the loss functions underlying statistical estimators to be fully separable into multiple components, so that we need a more flexible notion of separability, where different losses could be shown to be separable to differing degrees. In a very related note, though it might seem unrelated at first, the analysis of $\ell_\infty$ convergence rates of

statistical estimators considerably lags that of say $\ell_2$ rates (see for instance, the unified framework of [7], which is suited to $\ell_2$ rates but is highly sub-optimal for $\ell_\infty$ rates). In part, the analysis of $\ell_\infty$ rates is harder because it implicitly requires analysis at the level of individual coordinates of the parameter vector. While this is thus harder than an $\ell_2$ error analysis, intuitively this would be much easier if the loss function were to split into independent components involving individual coordinates. While general loss functions might not be so "fully separable", they might perhaps satisfy a softer notion of separability motivated above. In a contribution that would be of independent interest, we develop precisely such a softer notion of separability for general loss functions. We then use this notion of separability to derive $\ell_\infty$ convergence rates for general $M$-estimators.

Given this machinery, we are then able to contrast generative and discriminative models. We focus on the case where the generative models are specified by exponential family distributions, so that the corresponding discriminative models are logistic regression models with the generative model sufficient statistics as feature functions. To compare the convergence rates of the two models, we focus on the difference of the two generative model parameters, since this difference is also precisely the *model parameter for the discriminative model* counterpart of the generative model, via an application of the Bayes rule. Moreover, as Li et al. [3] and others show, the $\ell_2$ convergence rates of the difference of the two parameters is what drives the classification error rates of both generative as well as discriminative model classifiers. Incidentally, such a difference of generative model parameters has also attracted interest outside the context of classification, where it is called differential parameter learning [1, 14, 6]. We thus analyze the $\ell_\infty$ as well as $\ell_2$ rates for both the generative and discriminative models, focusing on this parameter difference. As we show, unlike the case of Naive Bayes and logistic regression in low-dimensions as studied in [8], this general high-dimensional setting is more nuanced, and in particular depends on the separability of the generative models. As we show, under some conditions on the models, generative and discriminative models not only have potentially different $\ell_\infty$ rates, but also differing "burn in" periods in terms of the minimum number of samples required in order for the convergence rates to hold. The choice of a generative vs discriminative model, namely that with a better sample complexity, thus depends on their corresponding separabilities. As a minor note, we also show how generative model $M$-estimators are not directly suitable in high-dimensions, and provide a simple methodological fix in order to obtain better $\ell_2$ rates. We instantiate our results with two running examples of isotropic and non-isotropic Gaussian generative models, and also corroborate our theory with instructive simulations.

## 2  Background and Setup.

We consider the problem of differential parameter estimation under the following generative model. Let $Y \in \{0, 1\}$ denote a binary response variable, and let $X = (X_1, \ldots, X_p) \in \mathbb{R}^p$ be the covariates. For simplicity, we assume $\mathbb{P}[Y = 1] = \mathbb{P}[Y = 0] = \frac{1}{2}$. We assume that conditioned on the response variable, the covariates belong to an exponential family, $X|Y \sim P_{\theta_Y^*}(\cdot)$, where:

$$\mathbb{P}_{\theta_Y^*}(X|Y) = h(X) \exp(\langle \theta_Y^*, \phi(X) \rangle - A(\theta_Y^*)). \tag{1}$$

Here, $\theta_Y^*$ is the vector of the true canonical parameters, $A(\theta)$ is the log-partition function and $\phi(X)$ is the sufficient statistic. We assume access to two sets of samples $\mathcal{X}_0^n = \{x_i^{(0)}\}_{i=1}^n \sim \mathbb{P}_{\theta_0^*}$ and $\mathcal{X}_1^n = \{x_i^{(1)}\}_{i=1}^n \sim \mathbb{P}_{\theta_1^*}$. Given these samples, as noted in the introduction, we are particularly interested in estimating the differential parameter $\theta_{\text{diff}}^* := \theta_1^* - \theta_0^*$, since this is also the model parameter corresponding to the discriminative model, as we show below. In high dimensional sampling settings, we additionally assume that $\theta_{\text{diff}}^*$ is at most $s$-sparse, *i.e.* $\|\theta_{\text{diff}}^*\|_0 \leq s$.

We will be using the following two exponential family generative models as running examples: isotropic and non-isotropic multivariate Gaussian models.

**Isotropic Gaussians (IG)**  Let $X = (X_1, \ldots, X_p) \sim \mathcal{N}(\mu, \mathcal{I}_p)$ be an isotropic gaussian random variable; it's density can be written as:

$$\mathbb{P}_\mu(x) = \frac{1}{\sqrt{(2\pi)^p}} \exp\left(-\frac{1}{2}(x - \mu)^{\mathrm{T}}(x - \mu)\right). \tag{2}$$

**Gaussian MRF (GMRF).**  Let $X = (X_1, \ldots, X_p)$ denote a zero-mean gaussian random vector; it's density is fully-parametrized as by the inverse covariance or concentration matrix $\Theta = (\Sigma)^{-1} \succ 0$

and can be written as:

$$\mathbb{P}_\Theta(x) = \frac{1}{\sqrt{(2\pi)^p \det\left((\Theta)^{-1}\right)}} \exp\left(-\frac{1}{2}x^{\mathrm{T}}\Theta x\right). \tag{3}$$

Let $d_\Theta = \max_{j\in[p]} \left\|\Theta_{(:,j)}\right\|_0$ is the maximum number non-zeros in a row of $\Theta$. Let $\kappa_{\Sigma^*} = \left\|\left\|(\Theta^*)^{-1}\right\|\right\|_\infty$, where $\left\|\left\|M\right\|\right\|_\infty$ is the $\ell_\infty/\ell_\infty$ operator norm given by $\left\|\left\|M\right\|\right\|_\infty = \max_{j=1,2,\ldots,p} \sum_{k=1}^p |M_{jk}|$.

**Generative Model Estimation.** Here, we proceed by estimating the two parameters $\{\theta_i^*\}_{i=0}^1$ individually. Letting $\widehat{\theta}_1$ and $\widehat{\theta}_0$ be the corresponding estimators, we can then estimate the difference of the parameters as $\widehat{\theta}_{\mathrm{diff}} = \widehat{\theta}_1 - \widehat{\theta}_0$. The most popular class of estimators for the individual parameters is based on Maximum likelihood Estimation (MLE), where we maximize the likelihood of the given data. For isotropic gaussians, the negative log-likelihood function can be written as:

$$\mathcal{L}_{n_{\mathrm{IG}}}(\theta) = \frac{\theta^T\theta}{2} - \theta^T\widehat{\mu}, \tag{4}$$

where $\widehat{\mu} = \frac{1}{n}\sum_{i=1}^n x_i$. In the case of GGMs the negative log-likelihood function can be written as:

$$\mathcal{L}_{n_{\mathrm{GGM}}}(\Theta) = \left\langle\left\langle\Theta, \widehat{\Sigma}\right\rangle\right\rangle - \log(\det(\Theta)), \tag{5}$$

where $\widehat{\Sigma} = \frac{1}{n}\sum_{i=1}^n x_i x_i^T$ is the sample covariance matrix and $\langle\langle U, V\rangle\rangle = \sum_{i,j} U_{ij}V_{ij}$ denotes the trace inner product on the space of symmetric matrices. In high-dimensional sampling regimes ($n << p$), regularized MLEs, for instance with $\ell_1$-regularization under the assumption of sparse model parameters, have been widely used [11, 10, 2].

**Discriminative Model Estimation.** Using Bayes rule, we have that:

$$\mathbb{P}[Y=1|X] = \frac{\mathbb{P}[X|Y=1]\mathbb{P}[Y=1]}{\mathbb{P}[X|Y=0]\mathbb{P}[Y=0] + \mathbb{P}[X|Y=1]\mathbb{P}[Y=1]}$$

$$= \frac{1}{1 + \exp\left(-\left(\langle\theta_1^* - \theta_0^*, \phi(x)\rangle + c^*\right)\right)} \tag{6}$$

where $c^* = A(\theta_0^*) - A(\theta_1^*)$. The conditional distribution is simply a logistic regression model, with the generative model sufficient statistics as the features, and with optimal parameters being precisely the difference $\theta_{\mathrm{diff}}^* := \theta_1^* - \theta_0^*$ of the generative model parameters. The corresponding negative log-likelihood function can be written as

$$\mathcal{L}_{\mathrm{logistic}}(\theta, c) = \frac{1}{n}\sum_{i=1}^n \left(-y_i(\langle\theta, \phi(x_i)\rangle + c) + \Phi(\langle\theta, \phi(x_i)\rangle + c)\right) \tag{7}$$

where $\Phi(t) = \log(1 + \exp(t))$. In high dimensional sampling regimes, under the assumption that the model parameters are sparse, we would use the $\ell_1$-penalized version $\widehat{\theta}_{\mathrm{diff}}$ of the MLE (7) to estimate $\theta_{\mathrm{diff}}^*$.

**Outline.** We proceed by studying the more general problem of $\ell_\infty$ error for parameter estimation for any loss function $\mathcal{L}_n(\cdot)$. Specifically, consider the general $M$-estimation problem, where we are given $n$ i.i.d samples $Z_1^n = \{z_1, z_2, \ldots, z_n\}, z_i \in \mathcal{Z}$ from some distribution $\mathbb{P}$, and we are interested in estimating some parameter $\theta^*$ of the distribution $\mathbb{P}$. Let $\ell : \mathbb{R}^p \times \mathcal{Z} \mapsto \mathbb{R}$ be a twice differentiable and convex function which assigns a loss $\ell(\theta; z)$ to any parameter $\theta \in \mathbb{R}^p$, for a given observation $z$. Also assume that the loss is Fisher consistent so that $\theta^* \in \mathrm{argmin}_\theta \bar{\mathcal{L}}(\theta)$ where $\bar{\mathcal{L}}(\theta) \stackrel{\mathrm{def}}{=} \mathbb{E}_{z\sim\mathbb{P}}[\ell(\theta; z)]$ is the population loss. We are then interested in analyzing the $M$-estimators $\theta^*$ that minimize the empirical loss *i.e.* $\widehat{\theta} \in \mathrm{argmin}_\theta \mathcal{L}_n(\theta)$, or regularized versions thereof, where $\mathcal{L}_n(\theta) = \frac{1}{n}\sum_{i=1}^n \mathcal{L}(\theta; Z_i)$.

We introduce a notion of the separability of a loss function, and show how more *separable* losses require fewer samples to establish convergence for $\left\|\widehat{\theta} - \theta^*\right\|_\infty$. We then instantiate our separability results from this general setting for both generative and discriminative models. We calculate the number of samples required for generative and discriminative approaches to estimate the differential parameter $\theta_{\mathrm{diff}}^*$, for consistent convergence rates with respect to $\ell_\infty$ and $\ell_2$ norm. We also discuss the consequences of these results for high dimensional classification for Gaussian Generative models.

# 3 Separability

Let $R(\Delta; \theta^*) = \nabla \mathcal{L}_n(\theta^* + \Delta) - \nabla \mathcal{L}_n(\theta^*) - \nabla^2 \mathcal{L}_n(\theta^*)\Delta$ be the error in the first order approximation of the gradient at $\theta^*$. Let $\mathcal{B}_\infty(r) = \{\theta | \|\theta\|_\infty \leq r\}$ be an $\ell_\infty$ ball of radius $r$. We begin by analyzing the low dimensional case, and then extend it to high dimensions.

## 3.1 Low Dimensional Sampling Regimes

In low dimensional sampling regimes, we assume that the number of samples $n \gg p$. In this setting, we make the standard assumption that the empirical loss function $\mathcal{L}_n(\cdot)$ is *strongly convex*. Let $\widehat{\theta} = \operatorname{argmin}_\theta \mathcal{L}_n(\theta)$ denote the unique minimizer of the empirical loss function. We begin by defining a notion of separability for any such empirical loss function $\mathcal{L}_n$.

**Definition 1.** $\mathcal{L}_n$ *is* $(\alpha, \beta, \gamma)$ *locally separable around* $\theta^*$ *if the remainder term* $R(\Delta; \theta^*)$ *satisfies:*

$$\|R(\Delta; \theta^*)\|_\infty \leq \frac{1}{\beta} \|\Delta\|_\infty^\alpha \quad \forall \Delta \in \mathcal{B}_\infty(\gamma)$$

This definition might seem a bit abstract, but for some general intuition, $\gamma$ indicates the region where it is separable, $\alpha$ indicates the conditioning of the loss, while it is $\beta$ that quantifies the degree of separability: the larger it is, the more separable the loss function. Next, we provide some additional intuition on how a loss function's separability is connected to $(\alpha, \beta, \gamma)$. Using the mean-value theorem, we can write $\|R(\Delta, \theta^*)\|_\infty = \left\| \left( \nabla^2 \mathcal{L}_n(\theta^* + t\Delta) - \nabla^2 \mathcal{L}_n(\theta^*) \right) \Delta \right\|_\infty$ for some $t \in (0, 1)$. This can be further simplified as $\|R(\Delta, \theta^*)\|_\infty \leq \left\| \nabla^2 \mathcal{L}_n(\theta^* + t\Delta) - \nabla^2 \mathcal{L}_n(\theta^*) \right\|_\infty \|\Delta\|_\infty$. Hence, $\alpha$ and $1/\beta$ measure the smoothness of Hessian (w.r.t. the $\ell_\infty/\ell_\infty$ matrix norm) in the neighborhood of $\theta^*$, with $\alpha$ being the smoothness exponent, and $1/\beta$ being the smoothness constant. Note that the Hessian of the loss function $\nabla^2 \mathcal{L}_n(\theta)$ is a random matrix and can vary from being a diagonal matrix for a fully-separable loss function to a dense matrix for a heavily-coupled loss function. Moreover, from standard concentration arguments, the $\ell_\infty/\ell_\infty$ matrix norm for a diagonal ("separable") subgaussian random matrix has at most logarithmic dimension dependence[1], but for a dense ("non-separable") random matrix, the $\ell_\infty/\ell_\infty$ matrix norm could possibly scale linearly in the dimension. Thus, the scaling of $\ell_\infty/\ell_\infty$ matrix norm gives us an indication how "separable" the matrix is. This intuition is captured by $(\alpha, \beta, \gamma)$, which we further elaborate in future sections by explicitly deriving $(\alpha, \beta, \gamma)$ for different loss functions and use them to derive $\ell_2$ and $\ell_\infty$ convergence rates.

**Theorem 1.** *Let* $\mathcal{L}_n$ *be a strongly convex loss function which is* $(\alpha, \beta, \gamma)$ *locally separable function around* $\theta^*$. *Then, if* $\|\nabla \mathcal{L}_n(\theta^*)\|_\infty \leq \min\{\frac{\gamma}{2\kappa}, \left(\frac{1}{2\kappa}\right)^{\frac{\alpha}{\alpha-1}} \beta^{\frac{1}{\alpha-1}}\}$

$$\left\|\widehat{\theta} - \theta^*\right\|_\infty \leq 2\kappa \|\nabla \mathcal{L}_n(\theta^*)\|_\infty$$

*where* $\kappa = \left\| \nabla^2 \mathcal{L}_n(\theta^*)^{-1} \right\|_\infty$.

*Proof.* (Proof Sketch). The proof begins by constructing a suitable continuous function $F$, for which $\widehat{\Delta} = \widehat{\theta} - \theta^*$ is the unique fixed point. Next, we show that $F(\mathcal{B}_\infty(r)) \subseteq \mathcal{B}_\infty(r)$ for $r = 2\kappa \|\nabla \mathcal{L}_n(\theta^*)\|_\infty$. Since $F$ is continuous and $\ell_\infty$-ball is convex and compact, the contraction property coupled with Brouwer's fixed point theorem [9], shows that there exists some fixed point $\Delta$ of $F$, such that $\|\Delta\|_\infty \leq 2\kappa \|\nabla \mathcal{L}_n(\theta^*)\|_\infty$. By uniqueness of the fixed point, we then establish our result. See Figure 1 for a geometric description and Section A for more details $\qquad \square$

## 3.2 High Dimensional Sampling Regimes

In high dimensional sampling regimes ($n << p$), estimation of model parameters is typically an under-determined problem. It is thus necessary to impose additional assumptions on the true model parameter $\theta^*$. We will focus on the popular assumption of sparsity, which entails that the number of non-zero coefficients of $\theta^*$ is small, so that $\|\theta^*\|_0 \leq s$. For this setting, we will be focusing in particular on $\ell_1$-regularized empirical loss minimization:

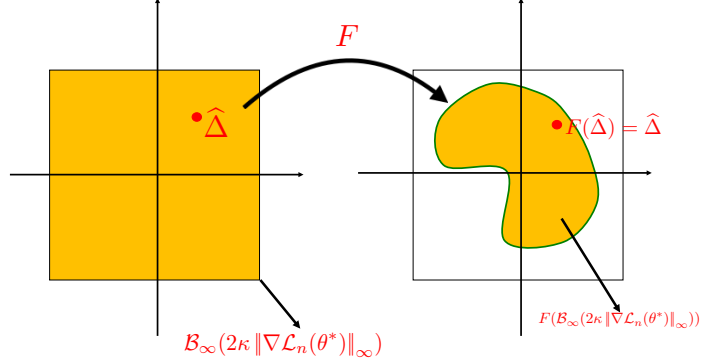

Figure 1: Under the conditions of Theorem 1, $F(\Delta) = -\nabla^2 \mathcal{L}_n(\theta^*)^{-1} \left( R(\Delta; \theta^*) + \nabla \mathcal{L}_n(\theta^*) \right)$ is contractive over $\mathcal{B}_\infty(2\kappa \|\nabla \mathcal{L}_n(\theta^*)\|_\infty)$ and has $\widehat{\Delta} = \widehat{\theta} - \theta^*$ as its unique fixed point. Using these two observations, we can conclude that $\left\| \widehat{\Delta} \right\|_\infty \leq 2\kappa \|\nabla \mathcal{L}_n(\theta^*)\|_\infty$.

$$\widehat{\theta}_{\lambda_n} = \operatorname*{argmin}_\theta \mathcal{L}_n(\theta) + \lambda_n \|\theta\|_1 \tag{8}$$

Let $S = \{i \mid \theta_i^* \neq 0\}$ be the support set of the true parameter and $\mathcal{M}(\mathcal{S}) = \{v | v_{S^c} = 0\}$ be the corresponding subspace. Note that under a high-dimensional sampling regime, we can no longer assume that the empirical loss $\mathcal{L}_n(\cdot)$ is *strongly convex*. Accordingly, we make the following set of assumptions:

- Assumption 1 (A1): **Positive Definite Restricted Hessian.** $\nabla^2_{SS} \mathcal{L}_n(\theta^*) \succsim \lambda_{\min} \mathcal{I}$
- Assumption 2 (A2): **Irrepresentability.** There exists some $\psi \in (0, 1]$ such that

$$\left\| \nabla^2_{S^c S} \mathcal{L}_n(\theta^*) \left( \nabla^2_{SS} \mathcal{L}_n(\theta^*) \right)^{-1} \right\|_\infty \leq 1 - \psi$$

- Assumption 3 (A3). **Unique Minimizer.** When restricted to the true support, the solution to the $\ell_1$ penalized loss minimization problem is unique, which we denote by:

$$\tilde{\theta}_{\lambda_n} = \operatorname*{argmin}_{\theta \in \mathcal{M}(S)} \left\{ \mathcal{L}_n(\theta) + \lambda_n \|\theta\|_1 \right\}. \tag{9}$$

Assumptions 1 and 2 are common in high dimensional analysis. We verify that Assumption 3 holds for different loss functions individually. We refer the reader to [13, 5, 11, 10] for further details on these assumptions. For this high dimensional sampling regime, we also modify our separability notion to a *restricted separability*, which entails that the remainder term be separable only over the model subspace $\mathcal{M}(S)$.

**Definition 2.** $\mathcal{L}_n$ *is* $(\alpha, \beta, \gamma)$ *restricted locally separable around* $\theta^*$ *over the subspace* $\mathcal{M}(\mathcal{S})$ *if the remainder term* $R(\Delta; \theta^*)$ *satisfies:*

$$\|R(\Delta; \theta^*)\|_\infty \leq \frac{1}{\beta} \|\Delta\|_\infty^\alpha \quad \forall \Delta \in \mathcal{B}_\infty(\gamma) \cap \mathcal{M}(S)$$

We present our main deterministic result in high dimensions.

**Theorem 2.** *Let* $\mathcal{L}_n$ *be a* $(\alpha, \beta, \gamma)$ *locally separable function around* $\theta^*$. *If* $(\lambda_n, \nabla \mathcal{L}_n(\theta^*))$ *are such that,*

- $\frac{\psi}{8} \lambda_n \geq \|\nabla \mathcal{L}_n(\theta^*)\|_\infty$.
- $\|\nabla \mathcal{L}_n(\theta^*)\|_\infty + \lambda_n \leq \min \left\{ \frac{\gamma}{2\kappa}, \left( \frac{1}{2\kappa} \right)^{\frac{\alpha}{\alpha-1}} \beta^{\frac{1}{\alpha-1}} \right\}$

*Then we have that* $support(\widehat{\theta}_{\lambda_n}) \subseteq support(\theta^*)$ *and*

$$\left\| \widehat{\theta}_{\lambda_n} - \theta^* \right\|_\infty \leq 2\kappa \left( \|\nabla \mathcal{L}_n(\theta^*)\|_\infty + \lambda_n \right)$$

*where* $\kappa = \left\| \nabla^2_{SS} \mathcal{L}_n(\theta^*)^{-1} \right\|_\infty$

*Proof.* (Proof Sketch). The proof invokes the primal-dual witness argument [13] which when combined with Assumption 1-3, gives $\widehat{\theta}_{\lambda_n} \in \mathcal{M}(S)$ and that $\widehat{\theta}_{\lambda_n}$ is the unique solution of the restricted problem. The rest of the proof proceeds similar to Theorem 1, by constructing a suitable function $F : \mathbb{R}^{|S|} \mapsto \mathbb{R}^{|S|}$ for which $\widehat{\Delta} = \widehat{\theta}_{\lambda_n} - \theta^*$ is the unique fixed point, and showing that $F$ is contractive over $\mathcal{B}_\infty(r; \theta^*)$ for $r = 2\kappa \left( \|\nabla \mathcal{L}_n(\theta^*)\|_\infty + \lambda_n \right)$. See Section B for more details. □

**Discussion.** Theorems 1 and 2 provide a general recipe to estimate the number of samples required by any loss $\ell(\theta, z)$ to establish $\ell_\infty$ convergence. The first step is to calculate the separability constants $(\alpha, \beta, \gamma)$ for the corresponding empirical loss function $\mathcal{L}_n$. Next, since the loss $\ell$ is Fisher consistent, so that $\nabla \bar{\mathcal{L}}(\theta^*) = 0$, the upper bound on $\|\nabla \mathcal{L}_n(\theta^*)\|_\infty$ can be shown to hold by analyzing the concentration of $\nabla \mathcal{L}_n(\theta^*)$ around its mean. We emphasize that we do not impose any restrictions on the values of $(\alpha, \beta, \gamma)$. In particular, these can scale with the number of samples $n$; our results hold so long as the number of samples $n$ satisfy the conditions of the theorem. As a rule of thumb, the smaller that either $\gamma$ or $\beta$ get for any given loss $\ell$, the larger the required number of samples.

# 4   $\ell_\infty$-rates for Generative and Discriminative Model Estimation

In this section we study the $\ell_\infty$ rates for differential parameter estimation for the discriminative and generative approaches. We do so by calculating the separability of discriminative and generative loss functions, and then instantiate our previously derived results.

## 4.1   Discriminative Estimation

As discussed before, the discriminative approach uses $\ell_1$-regularized logistic regression with the sufficient statistic as features to estimate the differential parameter. In addition to A1-A3, we assume column normalization of the sufficient statistics, *i.e.* $\sum_{i=1}^n \left( [\phi(x_i)]_j \right)^2 \leq n$. Let $\gamma_n = \max_i \|\phi(x)_i\|_\infty$, $\nu_n = \max_i \|(\phi(x)_i)_S\|_2$. Firstly, we characterize the separability of the logistic loss.

**Lemma 1.** *The logistic regression negative log-likelihood $\mathcal{L}_{n_{Logistic}}$ from (7) is $\left( 2, \frac{1}{s\gamma_n \nu_n^2}, \infty \right)$ restricted local separable around $\theta^*$.*

Combining Lemma 1 with Theorem 2, we get the following corollary.

**Corollary 3.** *(Logistic Regression) Consider the model in (1), then there exist universal positive constants $C_1, C_2$ and $C_3$ such that for $n \geq C_1 \kappa^2 s^2 \gamma_n^2 \nu_n^4 \log p$ and $\lambda_n = C_2 \sqrt{\frac{\log p}{n}}$, the discriminative differential estimate $\widehat{\theta}_{diff}$, satisfies*

$$support(\widehat{\theta}_{diff}) \subseteq support(\theta^*_{diff}) \;\; and \;\; \left\| \widehat{\theta}_{diff} - \theta^*_{diff} \right\|_\infty \leq C_3 \sqrt{\frac{\log p}{n}}.$$

## 4.2   Generative Estimation

We characterize the separability of Generative Exponential Families. The negative log-likelihood function can be written as:

$$\mathcal{L}_n(\theta) = A(\theta) - \langle \theta, \phi_n \rangle,$$

where $\phi_n = \frac{1}{n} \sum_{i=1}^n \phi(x_i)$. In this setting, the remainder term is independent of the data and can be written as $R(\Delta) = \nabla A(\theta^* + \Delta) - \nabla A(\theta^*) - \nabla^2 A(\theta^*)\Delta$ and $\nabla \mathcal{L}_n(\theta^*) = \mathbb{E}[\phi(x)] - \frac{1}{n}\phi(x_i)$. Hence, $\|\nabla \mathcal{L}_n(\theta^*)\|_\infty$ is a measure of how well the sufficient statistics concentrate around their mean. Next, we show the separability of our running examples Isotropic Gaussians and Gaussian Graphical Models.

**Lemma 2.** *The isotropic Gaussian negative log-likelihood $\mathcal{L}_{n_{IG}}$ from (4) is $(\cdot, \infty, \infty)$ locally separable around $\theta^*$.*

**Lemma 3.** *The Gaussian MRF negative log-likelihood $\mathcal{L}_{n_{GGM}}$ from (5) is $\left( 2, \frac{2}{3 d_\Theta^* \kappa_{\Sigma^*}^3}, \frac{1}{3 d_\Theta^* \kappa_{\Sigma^*}} \right)$ restricted locally separable around $\Theta^*$.*

Comparing Lemmas 1, 2 and 3, we see that the separability of the discriminative model loss depends weakly on the feature functions. On the other hand, the separability for the generative model loss depends critically on the underlying sufficient statistics. This has consequences for their differing sample complexities for differential parameter estimation, as we show next.

**Corollary 4.** *(Isotropic Gaussians) Consider the model in* (2). *Then there exist universal constants* $C_1, C_2, C_3$ *such that if the number of samples scale as* $n \geq C_1 \log p$, *then with probability atleast* $1 - 1/p^{C_2}$, *the generative estimate of the differential parameter* $\widehat{\theta}_{diff}$ *satisfies*

$$\left\| \widehat{\theta}_{diff} - \theta^*_{diff} \right\|_\infty \leq C_3 \sqrt{\frac{\log p}{n}}.$$

Comparing Corollary 3 and Corollary 4, we see that for isotropic gaussians, both the discriminative and generative approach achieve the same $\ell_\infty$ convergence rates, but at *different sample complexities*. Specifically, the sample complexity for the generative method depends only logarithmically on the dimension $p$, and is independent of the differential sparsity $s$, while the sample complexity of the discriminative method depends on the differential sparsity $s$. Therefore in this case, the generative method is strictly better than its discriminative counterpart, assuming that the generative model assumptions hold.

**Corollary 5.** *(Gaussian MRF) Consider the model in* (3)*, and suppose that the scaled covariates* $X_k / \sqrt{\Sigma^*_{kk}}$ *are subgaussian with parameter* $\sigma^2$. *Then there exist universal positive constants* $C_2, C_3, C_4$ *such that if the number of samples for the two generative models scale as* $n_i \geq C_2 \kappa_i^2 \kappa_{(\Theta^*_i)^{-1}}^6 d^2_{\Theta^*_i} \log p$, *for* $i \in \{0,1\}$, *then with probability at least* $1 - 1/p^{C_3}$, *the generative estimate of the differential parameter,* $\widehat{\Theta}_{diff} = \widehat{\Theta}_1 - \widehat{\Theta}_0$, *satisfies*

$$\left\| \widehat{\Theta}_{diff} - \Theta^*_{diff} \right\|_\infty \leq C_4 \sqrt{\frac{\log p}{n}},$$

*and* support$(\widehat{\Theta}_i) \subseteq$ support$(\Theta^*_i)$ *for* $i \in \{0,1\}$.

Comparing Corollary 3 and Corollary 5, we see that for Gaussian Graphical Models, both the discriminative and generative approach achieve the same $\ell_\infty$ convergence rates, but at different sample complexities. Specifically, the sample complexity for the generative method depends only on row-wise sparsity of the individual models $d^2_{\Theta^*_i}$, and is independent of sparsity $s$ of the differential parameter $\Theta^*_{\text{diff}}$. In contrast, the sample complexity of the discriminative method depends only on the sparsity of the differential parameter, and is independent of the structural complexities of the individual model parameters. This suggests that in high dimensions, even when the generative model assumptions hold, generative methods might perform poorly if the underlying model is highly non-separable (*e.g.* $d = \Omega(p)$), which is in contrast to the conventional wisdom in low dimensions.

**Related Work.** Note that results similar to Corollaries 3 and 5 have been previously reported in [11, 5] separately. Under the same set of assumptions as ours, Li et al. [5] provide a unified analysis for support recovery and $\ell_\infty$-bounds for $\ell_1$-regularized M-estimators. While they obtain the same rates as ours, their required sample complexities are much higher, since they do not exploit the separability of the underlying loss function. As one example, in the case of GMRFs, their results require the number of samples to scale as $n > k^2 \log p$, where $k$ is the total number of edges in the graph, which is sub-optimal, and in particular does not match the GMRF-specific analysis of [11]. On the other hand, our unified analysis is tighter, and in particular, does match the results of [11].

## 5 $\ell_2$-rates for Generative and Discriminative Model Estimation

In this section we study the $\ell_2$ rates for differential parameter estimation for the discriminative and generative approaches.

### 5.1 Discriminative Approach

The bounds for the discriminative approach are relatively straightforward. Corollary 3 gives bounds on the $\ell_\infty$ error and establishes that support$(\widehat{\theta}) \subseteq$ support$(\theta^*)$. Since the true model parameter is $s$-sparse, $\|\theta^*\|_0 \leq s$, the $\ell_2$ error can be simply bounded as $\sqrt{s} \|\widehat{\theta} - \theta^*\|_\infty$.

## 5.2 Generative Approach

In the previous section, we saw that the generative approach is able to exploit the inherent separability of the underlying model, and thus is able to get $\ell_\infty$ rates for differential parameter estimation at a much lower sample complexity. Unfortunately, it does not have support consistency. Hence a naïve generative estimator will have an $\ell_2$ error scaling with $\sqrt{\frac{p \log p}{n}}$, which in high dimensions, would make it unappealing. However, one can exploit the sparsity of $\theta^*_{\text{diff}}$ and get better rates of convergence in $\ell_2$-norm by simply soft-thresholding the generative estimate. Moreover, soft-thresholding also leads to support consistency.

**Definition 3.** *We denote the soft-thresholding operator $ST_{\lambda_n}(\cdot)$, defined as:*

$$ST_{\lambda_n}(\theta) = \underset{w}{\arg\min}\, \frac{1}{2}\|w - \theta\|_2^2 + \lambda_n \|w\|_1.$$

**Lemma 4.** *Suppose $\theta = \theta^* + \epsilon$ for some $s$-sparse $\theta^*$. Then there exists a universal constant $C_1$ such that for $\lambda_n \geq 2\|\epsilon\|_\infty$,*

$$\|ST_{\lambda_n}(\theta) - \theta^*\|_2 \leq C_1\sqrt{s}\|\epsilon\|_\infty \quad and \quad \|ST_{\lambda_n}(\theta) - \theta^*\|_1 \leq C_1 s \|\epsilon\|_\infty \tag{10}$$

Note that this is a completely deterministic result and has no sample complexity requirement. Motivated by this, we introduce a thresholded generative estimator that has two stages: (a) compute $\widehat{\theta}_{\text{diff}}$ using the generative model estimates, and (b) soft-threshold the generative estimate with $\lambda_n = c\left\|\widehat{\theta}_{\text{diff}} - \theta^*_{\text{diff}}\right\|_\infty$. An elementary application of Lemma 4 can then be shown to provide $\ell_2$ error bounds for $\widehat{\theta}_{\text{diff}}$ given its $\ell_\infty$ error bounds, and that the true parameter $\theta^*_{\text{diff}}$ is $s$-sparse. We instantiate these $\ell_2$-bounds via corollaries for our running examples of Isotropic Gaussians, and Gaussian MRFs.

**Lemma 5.** *(Isotropic Gaussians) Consider the model in (2). Then there exist universal constants $C_1, C_2, C_3$ such that if the number of samples scale as $n \geq C_1 \log p$, then with probability atleast $1 - 1/p^{C_2}$, the soft-thresholded generative estimate of the differential parameter $ST_{\lambda_n}\left(\widehat{\theta}_{\text{diff}}\right)$, with the soft-thresholding parameter set as $\lambda_n = c\sqrt{\frac{\log p}{n}}$ for some constant $c$, satisfies:*

$$\left\|ST_{\lambda_n}\left(\widehat{\theta}_{\text{diff}}\right) - \theta^*_{\text{diff}}\right\|_2 \leq C_3\sqrt{\frac{s \log p}{n}}.$$

**Lemma 6.** *(Gaussian MRF) Consider the model in Equation 3, and suppose that the covariates $X_k/\sqrt{\Sigma^*_{kk}}$ are subgaussian with parameter $\sigma^2$. Then there exist universal positive constants $C_2, C_3, C_4$ such that if the number of samples for the two generative models scale as $n_i \geq C_2 \kappa_i^2 \kappa_{(\Theta_i^*)^{-1}}^6 d_{\Theta_i^*}^2 \log p$, for $i \in \{0, 1\}$, for $i \in \{0, 1\}$, then with probability at least $1 - 1/p^{C_3}$, the soft-thresholded generative estimate of the differential parameter, $ST_{\lambda_n}\left(\widehat{\Theta}_{\text{diff}}\right)$, with the soft-thresholding parameter set as $\lambda_n = c\sqrt{\frac{\log p}{n}}$ for some constant $c$, satisfies:*

$$\left\|ST_{\lambda_n}\left(\widehat{\Theta}_{\text{diff}}\right) - \Theta^*_{\text{diff}}\right\|_2 \leq C_4\sqrt{\frac{s \log p}{n}}.$$

Comparing Lemmas 5 and 6 to Section 5.1, we can see that the additional soft-thresholding step allows the generative methods to achieve the same $\ell_2$-error rates as the discriminative methods, but at different sample complexities. The sample complexities of the generative estimates depend on the separabilities of the individual models, and is independent of the differential sparsity $s$, where as the sample complexity of the discriminative estimate depends only on the differential sparsity $s$.

## 6 Experiments: High Dimensional Classification

In this section, we corroborate our theoretical results on $\ell_2$-error rates for generative and discriminative model estimators, via their consequences for high dimensional classification. We focus on the case of isotropic Gaussian generative models $X|Y \sim \mathcal{N}(\mu_Y, \mathcal{I}_p)$, where $\mu_0, \mu_1 \in \mathbb{R}^p$ are unknown

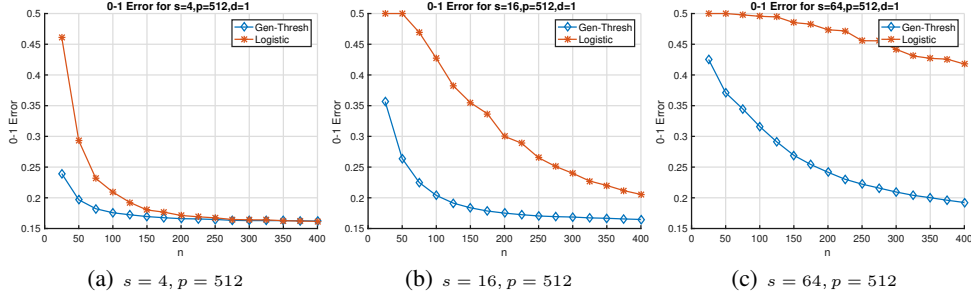

|(a) $s = 4, p = 512$|(b) $s = 16, p = 512$|(c) $s = 64, p = 512$|

Figure 2: Effect of sparsity $s$ on excess $0 - 1$ error.

and $\mu_1 - \mu_0$ is $s$-sparse. Here, we are interested in a classifier $C : \mathbb{R}^p \mapsto \{0, 1\}$ that achieves low classification error $\mathbb{E}_{X,Y}\left[\mathbf{1}\left\{C(X) \neq Y\right\}\right]$. Under this setting, it can be shown that the *Bayes classifier*, that achieves the lowest possible classification error, is given by the linear discriminant classifier $C^*(x) = \mathbf{1}\left\{x^T w^* + b^* > 0\right\}$, where $w^* = (\mu_1 - \mu_0)$ and $b^* = \frac{\mu_0^T \mu_0 - \mu_1^T \mu_1}{2}$. Thus, the coefficient $w^*$ of the linear discriminant is precisely the differential parameter, which can be estimated via both generative and discriminative approaches as detailed in the previous section. Moreover, the classification error can also be related to the $\ell_2$ error of the estimates. Under some mild assumptions, Li et al. [3] showed that for any linear classifier $\widehat{C}(x) = \mathbf{1}\left\{x^T \widehat{w} + \widehat{b} > 0\right\}$, the excess classification error can be bounded as:

$$\mathcal{E}(\widehat{C}) \leq C_1 \left( \|\widehat{w} - w^*\|_2^2 + \left\|\widehat{b} - b^*\right\|_2^2 \right),$$

for some constant $C_1 > 0$, and where $\mathcal{E}(C) = \mathbb{E}_{X,Y}\left[\mathbf{1}\left\{C(X) \neq Y\right\}\right] - \mathbb{E}_{X,Y}\left[\mathbf{1}\left\{C^*(X) \neq Y\right\}\right]$ is the excess 0-1 error. In other words, the excess classification error is bounded by a constant times the $\ell_2$ error of the differential parameter estimate.

**Methods.** In this setting, as discussed in previous sections, the discriminative model is simply a logistic regression model with linear features (6), so that the discriminative estimate of the differential parameter $\widehat{w}$ as well as the constant bias term $\widehat{b}$ can be simply obtained via $\ell_1$-regularized logistic regression. For the generative estimate, we use our two stage estimator from Section 5, which proceeds by estimating $\widehat{\mu}_0, \widehat{\mu}_1$ using the empirical means, and then estimating the differential parameter by soft-thresholding the difference of the generative model parameter estimates $\widehat{w}_T = \mathrm{ST}_{\lambda_n}\left(\widehat{\mu}_1 - \widehat{\mu}_0\right)$ where $\lambda_n = C_1 \sqrt{\frac{\log p}{n}}$ for some constant $C_1$. The corresponding estimate for $b^*$ is given by $\hat{b}_T = -\frac{1}{2}\left\langle \widehat{w}_T, \widehat{\mu}_1 + \widehat{\mu}_0 \right\rangle$.

**Experimental Setup.** For our experimental setup, we consider isotropic Gaussian models with means $\mu_0 = 1_p - \frac{1}{\sqrt{s}}\begin{bmatrix} 1_s \\ 0_{p-s} \end{bmatrix}, \mu_1 = 1_p + \frac{1}{\sqrt{s}}\begin{bmatrix} 1_s \\ 0_{p-s} \end{bmatrix}$, and vary the sparsity level $s$. For both methods, we set the regularization parameter [2] as $\lambda_n = \sqrt{\log(p)/n}$. We report the excess classification error for the two approaches, averaged over 20 trials, in Figure 2.

**Results.** As can be seen from Figure 2, our two-staged thresholded generative estimator is always better than the discriminative estimator, across different sparsity levels $s$. Moreover, the sample complexity or "burn-in" period of the discriminative classifier strongly depends on the sparsity level, which makes it unsuitable when the true parameter is not highly sparse. For our two-staged generative estimator, we see that the sparsity $s$ has no effect on the "burn-in" period of the classifier. These observations validate our theoretical results from Section 5.

## Acknowledgements

A.P. and P.R. acknowledge the support of ARO via W911NF-12-1-0390 and NSF via IIS-1149803, IIS-1447574, DMS-1264033, and NIH via R01 GM117594-01 as part of the Joint DMS/NIGMS Initiative to Support Research at the Interface of the Biological and Mathematical Sciences.

## Footnotes

[1]Follows from the concentration of subgaussian maxima [12]

[2]See Appendix J for cross-validated plots.

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
