[Supplementary Material · supp.pdf]

# A Proof of Theorem 1

*Proof.* The proof involves two steps:

- Constructing a suitable function $F : \mathbb{R}^p \mapsto \mathbb{R}^p$ for which the $\widehat{\theta}$ is the unique fixed point.(See Lemma 7)
- Showing that the function is contractive over an $\ell_\infty$ ball of radius $2\kappa \left\| \nabla L(\theta^*) \right\|_\infty$.

**Lemma 7.** *Let $\widehat{\theta} = \operatorname{argmin}_\theta \mathcal{L}_n(\theta)$. Then $\widehat{\Delta} = \widehat{\theta} - \theta^*$ is the unique fixed point of $F : \mathbb{R}^p \mapsto \mathbb{R}^p$,*

$$F(\Delta) = -\nabla^2 \mathcal{L}_n(\theta^*)^{-1} R(\Delta; \theta^*) - \nabla^2 \mathcal{L}_n(\theta^*)^{-1} \nabla \mathcal{L}_n(\theta^*)$$

*Proof.* By first order optimality of $\widehat{\theta}$, we know that $\nabla \mathcal{L}_n(\widehat{\theta}) = 0$. Using invertibility of $\nabla^2 \mathcal{L}_n(\theta^*)$, we know that for $F(\Delta) = \Delta - \nabla^2 \mathcal{L}_n(\theta^*)^{-1} \left( \nabla \mathcal{L}_n(\theta^* + \Delta) \right)$, $\widehat{\Delta}$ is the unique fixed point. Now, we can simplify $F$ as:

$$
\begin{aligned}
F(\Delta) &= \Delta - \nabla^2 \mathcal{L}_n(\theta^*)^{-1} \left( \nabla \mathcal{L}_n(\theta^* + \Delta) \right) \\
&= \Delta - \nabla^2 \mathcal{L}_n(\theta^*)^{-1} \left( \nabla \mathcal{L}_n(\theta^* + \Delta) \right) \\
&= \Delta - \nabla^2 \mathcal{L}_n(\theta^*)^{-1} \left( \nabla \mathcal{L}_n(\theta^* + \Delta) - \nabla \mathcal{L}_n(\theta^*) + \nabla \mathcal{L}_n(\theta^*) \right) \\
&= \Delta - \nabla^2 \mathcal{L}_n(\theta^*)^{-1} \left( R(\Delta; \theta^*) + \nabla^2 \mathcal{L}_n(\theta^*) \Delta + \nabla \mathcal{L}_n(\theta^*) \right) \\
&= -\nabla^2 \mathcal{L}_n(\theta^*)^{-1} R(\Delta; \theta^*) - \nabla^2 \mathcal{L}_n(\theta^*)^{-1} \nabla \mathcal{L}_n(\theta^*)
\end{aligned}
$$

where by definition, we have that $R(\Delta; \theta^*) = \nabla \mathcal{L}_n(\theta^* + \Delta) - \nabla \mathcal{L}_n(\theta^*) - \nabla^2 \mathcal{L}_n(\theta^*)$. $\qquad\square$

Now to show contraction of $F$. Let $r = 2\kappa \left\| \nabla \mathcal{L}_n(\theta^*) \right\|_\infty$ and let $\Delta \in \mathcal{B}_\infty(r)$

$$
\begin{aligned}
\left\| F(\Delta) \right\|_\infty &= \left\| -(\nabla^2 \mathcal{L}_n(\theta^*))^{-1} R(\Delta; \theta^*) - (\nabla^2 \mathcal{L}_n(\theta^*))^{-1} \nabla \mathcal{L}_n(\theta^*) \right\|_\infty \\
&\leq \kappa \left\| R(\Delta; \theta^*) \right\|_\infty + \kappa \left\| \nabla \mathcal{L}_n(\theta^*) \right\|_\infty
\end{aligned}
$$

By our assumption, $r = 2\kappa \left\| \nabla \mathcal{L}_n(\theta^*) \right\|_\infty \leq \gamma$, so we can upper bound $R(\Delta; \theta^*)$ using the separability of $\mathcal{L}_n$.

$$
\begin{aligned}
&\leq \kappa \frac{1}{\beta} (2\kappa \left\| \nabla \mathcal{L}_n(\theta^*) \right\|_\infty)^\alpha + r/2 \\
&= \frac{\kappa}{\beta} (2\kappa)^\alpha \left\| \nabla \mathcal{L}_n(\theta^*) \right\|_\infty^\alpha \times \frac{\kappa \left\| \nabla \mathcal{L}_n(\theta^*) \right\|_\infty}{\kappa \left\| \nabla \mathcal{L}_n(\theta^*) \right\|_\infty} + r/2 \\
&= \frac{1}{\beta} (2\kappa)^\alpha \left\| \nabla \mathcal{L}_n(\theta^*) \right\|_\infty^{\alpha-1} \times \kappa \left\| \nabla \mathcal{L}_n(\theta^*) \right\|_\infty + r/2 \\
&\leq \kappa \left\| \nabla \mathcal{L}_n(\theta^*) \right\|_\infty + r/2 \\
&\leq r
\end{aligned}
$$

where the last step follows from our assumption that $\left\| \nabla \mathcal{L}_n(\theta^*) \right\|_\infty \leq \left( \frac{1}{2\kappa} \right)^{\frac{\alpha}{\alpha-1}} \beta^{\frac{1}{\alpha-1}}$.
Hence, we've shown that $F(\mathcal{B}_\infty(r)) \subseteq \mathcal{B}_\infty(r)$ for $r = 2\kappa \left\| \nabla \mathcal{L}_n(\theta^*) \right\|_\infty$.

Since $F$ is continuous and $\ell_\infty$-ball is convex and compact, the contraction property coupled with Brouwer's fixed point theorem [9], shows that there exists some fixed point $\Delta$ of $F$, such that $\left\| \Delta \right\|_\infty \leq 2\kappa \left\| \nabla \mathcal{L}_n \right\|_\infty$. Lemma 7 established that $\widehat{\Delta} = \widehat{\theta} - \theta^*$ as the unique fixed point, hence, $\left\| \widehat{\Delta} \right\|_\infty \leq 2\kappa \left\| \nabla \mathcal{L}_n \right\|_\infty$. $\qquad\square$

# B Proof of Theorem 2

The proof involves three steps:

- Firstly, in Lemma 8, using the primal-dual witness argument of Wainwright [13], under Assumptions 1-3, we establish that the minimizer of the unconstrained problem(8) is unique and is equal to that of the restricted problem(9).( $\widehat{\theta}_{\lambda_n} = \widetilde{\theta}_{\lambda_n}$ )

- Using our assumption that the minimizer of the restricted function is unique, if we take partial derivatives of the Lagrangian of the restricted problem(9) with respect to the unconstrained elements, these partial derivatives must vanish at the optimum, meaning that we have the zero-gradient condition

$$G(\theta_S) = [\nabla \mathcal{L}_n(\theta)]_S + \lambda \tilde{Z}_S = 0 \tag{11}$$

where $\theta \in \mathbb{R}^p$ is such that, it's entries in $S$ is equal to $\theta_S$ and $0$ in $S^c$. The zero-gradient condition is necessary and sufficient for an optimum of the Lagrangian problem, and has a unique solution, $[\tilde{\theta}_{\lambda_n}]_S$.

$$G(\theta_S) = 0 \iff \theta_S = [\tilde{\theta}_{\lambda_n}]_S$$

Using this and following the proof of Theorem A, we construct a function $F : \mathbb{R}^{|S|} \mapsto \mathbb{R}^{|S|}$,

$$F(\Delta_S) = \Delta_S - (\nabla_{SS}^2 \mathcal{L}_n(\theta^*))^{-1} (G(\theta_S^* + \Delta_S)) \tag{12}$$

Since $\nabla_{SS}^2 \mathcal{L}_n(\theta^*)$ is invertible, $[\tilde{\Delta}]_S = [\tilde{\theta}_{\lambda_n} - \theta^*]_S$ is the unique optimal. $F$ can be simplified and written as:

$$F(\Delta_S) = \Delta_S - (\nabla_{SS}^2 \mathcal{L}_n(\theta^*))^{-1} \left[ R(\Delta) + \nabla^2 \mathcal{L}(\theta^*)\Delta + \nabla \mathcal{L}_n(\theta^*) \right]_S - (\nabla_{SS}^2 \mathcal{L}_n(\theta^*))^{-1} \lambda_n \tilde{Z}_S$$

where $\Delta \in \mathbb{R}^p$ is such that, it's entries in $S$ are equal to $\Delta_S$ and $0$ in $S^c$. Following some algebra, we can show that:

$$F(\Delta_S) = -(\nabla_{SS}^2 \mathcal{L}_n(\theta^*))^{-1} [R(\Delta)]_S - (\nabla_{SS}^2 \mathcal{L}_n(\theta^*))^{-1} [\nabla \mathcal{L}_n(\theta^*)]_S - \lambda_n (\nabla_{SS}^2 \mathcal{L}_n(\theta^*))^{-1} \tilde{Z}_S$$

- Note that $F(\cdot)$ is continuous. We'll show that $F$ is contractive over $\mathcal{B}_\infty(r)$ for $r = 2\kappa(\|\nabla \mathcal{L}_n(\theta^*)\|_\infty + \lambda_n)$. Let $\Delta_S \in \mathcal{B}_\infty(r)$. Then, the corresponding $\Delta \in \mathbb{R}^p$ is such that, it's entries in $S$ are equal to $\Delta_S$ and $0$ in $S^c$. Note that this $\Delta \in \mathcal{B}_\infty(r) \cap \mathcal{M}(S)$, and by assumption $r = 2\kappa(\|\nabla \mathcal{L}_n(\theta^*)\|_\infty + \lambda_n) \leq \gamma$, hence we can upper bound $\|R(\Delta)\|_\infty$ using that $\mathcal{L}_n$ is restricted locally separable around $\theta^*$ for this radius $r$.

$$\|F(\Delta_S)\|_\infty \leq \left\| -(\nabla_{SS}^2 \mathcal{L}_n(\theta^*))^{-1} [R(\Delta)]_S - (\nabla_{SS}^2 \mathcal{L}_n(\theta^*))^{-1} [\nabla \mathcal{L}_n(\theta^*)]_S - \lambda_n (\nabla_{SS}^2 \mathcal{L}_n(\theta^*))^{-1} \tilde{Z}_S \right\|_\infty$$

$$\tag{13}$$

$$\leq \kappa \|[R(\Delta)]_S\|_\infty + \kappa \left( \|[\nabla \mathcal{L}_n(\theta^*)]_S\|_\infty + \lambda_n \right) \tag{14}$$

$$\leq \kappa \|[R(\Delta)]_S\|_\infty + \kappa \left( \|[\nabla \mathcal{L}_n(\theta^*)]_S\|_\infty + \lambda_n \right) \tag{15}$$

$$\leq \kappa \underbrace{\|R(\Delta)\|_\infty}_{\leq \frac{1}{\beta} \|\Delta\|_\infty^\alpha} + \underbrace{\kappa \left( \|\nabla \mathcal{L}_n(\theta^*)\|_\infty + \lambda_n \right)}_{r/2} \tag{16}$$

$$\leq \kappa/\beta (2\kappa)^\alpha \left( \|\nabla \mathcal{L}_n(\theta^*)\|_\infty + \lambda_n \right)^\alpha + r/2 \tag{17}$$

$$\leq \underbrace{\left( \|\nabla \mathcal{L}_n(\theta^*)\|_\infty + \lambda_n \right)}_{r/2} + r/2 \tag{18}$$

$$\leq r \tag{19}$$

where we've used that by assumption, $\left( \|\nabla \mathcal{L}_n(\theta^*)\|_\infty + \lambda_n \right)^{\alpha - 1} \leq \frac{1}{2\kappa} \beta$.

**Lemma 8.** *Let $\frac{\alpha \lambda_n}{8} \geq \max\{\|\nabla \mathcal{L}_n(\theta^*)\|_\infty, \|R(\Delta)\|_\infty\}$. Then, $\widehat{\theta}_{\lambda_n} = \tilde{\theta}_{\lambda_n}$ is the unique minimizer of Equation 8. i.e. $Support(\widehat{\theta}_{\lambda_n}) \subseteq Support(\theta^*)$.*

*Proof.* The proof follows from the primal-dual witness argument of [13, 11]. We provide it here completeness. The steps are outlined as:

1. By assumption we have that $\tilde{\theta}_{\lambda_n}$ is the unique minimizer of the restricted problem.

$$\tilde{\theta}_{\lambda_n} = \operatorname*{argmin}_{\theta \in \mathcal{M}(S)} \mathcal{L}_n(\theta) + \lambda_n \|\theta\|_1 \tag{20}$$

2. Choose $\tilde{Z}$ such that $\tilde{Z}_S$ is sub-differential of $\left\| \tilde{\theta}_{\lambda_n} \right\|_1$.

3. Choose $\tilde{Z}_{S^c}$ such that $\nabla \mathcal{L}_n(\tilde{\theta}_{\lambda_n}) + \lambda \tilde{Z} = 0$, which ensures that $(\tilde{\theta}_{\lambda_n}, \tilde{Z})$ satisfy the optimality condition for (unconstrained) problem.(8).

4. Verify the strict dual feasibility condition

$$\left\| \tilde{Z}_{S^c} \right\|_\infty < 1$$

Steps 1-3 ensure that $(\tilde{\theta}_{\lambda_n}, \tilde{Z})$ satisfy the optimality conditions of (8). By construction, step 2 ensures that $\tilde{Z}_S$ satisfies the sub-differential conditions. Then, Step 4 is needed to ensure that the remaining elements of $\tilde{Z}$ satisfy the sub-differential conditions. If steps 1-4, succeed, then, it acts as a witness that the solution $\tilde{\theta}_{\lambda_n}$ to restricted problem is equal to solution of the unrestricted(original) problem.(8).

We now show the uniqueness of $\widehat{\theta}_{\lambda_n}$. By following an argument similar to Lemma 1 in [10],(Lemma 1(b) in [13]), any minimizer $\widehat{\theta}_{\lambda_n}$ of the original optimization problem satisfies $[\widehat{\theta}_{\lambda_n}]_{S^c} = 0$. Thus, since $\tilde{\theta}_{\lambda_n}$ is the only optimal vector for the restricted optimization problem(by assumption), we conclude that $\widehat{\theta}_{\lambda_n} = \tilde{\theta}_{\lambda_n}$.

Now, to show that $\left\| \tilde{Z}_{S^c} \right\|_\infty <$. Let $\tilde{\Delta} = \tilde{\theta}_{\lambda_n} - \theta^*$. So, we know $\tilde{\Delta}_{S^c} = 0$. Using this we can write the gradient condition. For brevity, let $\tilde{\theta} = \tilde{\theta}_{\lambda_n}$, $\nabla^2 \mathcal{L}_n(\theta^*) = \Gamma$.

$$\nabla \mathcal{L}_n(\tilde{\theta}) + \lambda_n \tilde{Z} = 0 \tag{21}$$

We can rewrite the gradient:

$$\Gamma_{SS} \tilde{\Delta} + [R(\tilde{\Delta})]_S + (\nabla \mathcal{L}_n(\theta^*))_S + \lambda_n \tilde{Z}_S = 0 \tag{22}$$
$$\Gamma_{S^c S} \tilde{\Delta} + [R(\tilde{\Delta})]_{S^c} + (\nabla \mathcal{L}_n(\theta^*))_{S^c} + \lambda_n \tilde{Z}_{S^c} = 0 \tag{23}$$

Since, $\Gamma_{SS}$ is invertible,

$$\tilde{\Delta} = \Gamma_{SS}^{-1} \left[ [R(\tilde{\Delta})]_S + (\nabla \mathcal{L}_n(\theta^*))_S + \lambda_n \tilde{Z}_S \right] \tag{24}$$

Plugging this into $\tilde{Z}_{S^c}$,

$$\left\| \tilde{Z}_{S^c} \right\|_\infty = \frac{1}{\lambda_n} \left\| [R(\tilde{\Delta})]_{S^c} + (\nabla \mathcal{L}_n(\theta^*))_{S^c} + \Gamma_{S^c S}(\Gamma_{SS})^{-1} \left[ [R(\tilde{\Delta})]_S + (\nabla \mathcal{L}_n(\theta^*))_S + \lambda_n \tilde{Z}_S \right] \right\|_\infty \tag{25}$$

$$\leq \frac{2 - \psi}{\lambda_n} \left( \left\| [R(\tilde{\Delta})] \right\|_\infty + \left\| (\nabla \mathcal{L}_n(\theta^*)) \right\|_\infty \right) + (1 - \psi) \tag{26}$$

$$\leq \frac{2 - \psi}{\lambda_n} \psi \lambda_n / 4 + (1 - \psi) \tag{27}$$

$$\leq 1 - \psi/2 < 1 \tag{28}$$

where we've used that $\left\| \Gamma_{S^c S}(\Gamma_{SS})^{-1} \right\|_\infty \leq 1 - \psi$, and $\left\| \tilde{Z}_S \right\|_\infty \leq 1$ in the first step. The next step follows from our assumption that $\max\{ \left\| \nabla \mathcal{L}_n(\theta^*) \right\|_\infty, \left\| R(\Delta) \right\|_\infty \} \leq \psi\lambda/8$. $\qquad\square$

## C  Proof of Lemma 1

*Proof.* The proof follows the analysis of [10]. Let $\mathcal{L}_n$ be the logistic loss. Let $\Delta \in \mathcal{B}_\infty(\gamma) \cap \mathcal{M}(S)$. For the logistic loss, the remainder term can be written as:

$$R(\Delta) = \nabla \mathcal{L}_n(\theta^* + \Delta) - \nabla \mathcal{L}_n(\theta^*) - \nabla^2 \mathcal{L}_n(\theta^*)\Delta \tag{29}$$
$$= \left[ \nabla^2 \mathcal{L}_n(\bar{\theta}) - \nabla^2 \mathcal{L}_n(\theta^*) \right] \Delta \tag{30}$$

where from mean-value theorem $\bar{\theta} = t(\theta^* + \Delta) + (1 - t)\theta^*$ is point on line joining $\theta^* + \Delta$ and $\theta^*$. Now, looking at the $j^{th}$ entry, we get:

$$[R(\Delta)]_j = \left[\nabla^2 \mathcal{L}_n(\bar{\theta}) - \nabla^2 \mathcal{L}_n(\theta^*)\right]_j \Delta \tag{31}$$

$$= \left[\frac{1}{n}\sum_{i=1}^{n} x_i x_i^T (\eta(\bar{\theta}^T x_i) - \eta(\theta^{*T} x_i))\right]_j \Delta \tag{32}$$

$$= \frac{1}{n}\sum_{i=1}^{n} (\eta(\bar{\theta}^T x_i) - \eta(\theta^{*T} x_i)) x_i^{(j)} x_i^T \Delta \tag{33}$$

where $\eta(t) = \frac{exp(t)}{(1+exp(t))^2}$. Again applying, mean-value theorem, where $\bar{\bar{\theta}}$ is another point between $(\theta^* + \Delta)$ and $\theta^*$, we get that

$$[R(\Delta)]_j = \frac{1}{n}\sum_{i=1}^{n} \eta'(\bar{\bar{\theta}}^T x_i)(x_i^T(\bar{\theta} - \theta^*)) x_i^{(j)} x_i^T \Delta \tag{34}$$

$$= \frac{1}{n}\sum_{i=1}^{n} \underbrace{\left[\eta'(\bar{\bar{\theta}}^T x_i) x_i^{(j)}\right]}_{a_i} \underbrace{\left[(\bar{\theta} - \theta^*)^T x_i x_i^T \Delta\right]}_{b_i} \tag{35}$$

$$= \frac{1}{n}\sum_{i=1}^{n} a_i b_i \tag{36}$$

By holder's inequality

$$|[R(\Delta)]_j| = \frac{1}{n}\left|\sum_{i=1}^{n} a_i b_i\right| \leq \frac{1}{n}\|a\|_\infty \|b\|_1 \tag{37}$$

Now $\|a\|_\infty \leq \gamma_n$. Also, observe that by definition $\bar{\theta} - \theta^* = t\Delta$

$$\frac{1}{n}\|b\|_1 = t\Delta^T \left\{\frac{1}{n}\sum_{i=1}^{n} x_i x_i^T\right\} \Delta \tag{38}$$

$$= t\Delta_S^T \left\{\frac{1}{n}\sum_{i=1}^{n} (x_i)_S (x_i)_S^T\right\} \Delta_S \tag{39}$$

$$\leq \nu_n^2 \|\Delta_S\|_2^2 \tag{40}$$

$$\leq \nu_n^2 s \|\Delta\|_\infty^2 \tag{41}$$

where we've used that, $\Delta \in \mathcal{M}(S)$. Combining the above, we get that,

$$\|R(\Delta)\|_\infty \leq \gamma_n \nu_n^2 s \|\Delta\|_\infty^2$$

$\square$

Hence, the logistic loss is $(2, \frac{1}{\gamma_n \nu_n^2 s}, \infty)$ separable around $\theta^*$.

## D  Proof of Corollary 3

In Lemma 1, we've already established the separability of logistic loss. Assume $\lambda_n = c. \|\nabla \mathcal{L}_n(\theta^*)\|_\infty$. We need to control $\left\|\nabla \mathcal{L}_{n_{\text{logistic}}}(\theta^*)\right\|_\infty$, so that $\|\nabla \mathcal{L}_n(\theta^*)\|_\infty <= \min\left\{\frac{\gamma}{2\kappa}, \left(\frac{1}{2\kappa}\right)^{\frac{\alpha}{\alpha-1}} \beta^{\frac{1}{\alpha-1}}\right\}$.

*Proof.*

$$\mathcal{L}_{n_{\text{logistic}}}(\theta) = \frac{1}{n}\sum_{i=1}^{n} \left[-y_i(\theta^T \phi(x_i)) + \log(1 + \exp(\langle \theta, \phi(x_i)\rangle)\right] \tag{42}$$

$$\mathcal{L}_{n_{\text{logistic}}}(\theta) = \frac{1}{n}\sum_{i=1}^{n} \phi(x_i)\left[-y_i + u(\theta^T \phi(x_i))\right], \tag{43}$$

where $u(t) = \frac{1}{1+\exp(-t)}$. Note that $u(\theta^{*T}\phi(x_i)) = \mathbb{P}(Y_i = 1|\phi(x_i)) = \mathbb{E}(Y_i)$
The jth-coordinate of $\mathcal{L}_{n_{\text{logistic}}}(\theta^*)$ can then be written as:

$$[\nabla \mathcal{L}_{n_{\text{Logistic}}}(\theta^*)]_j = -\frac{1}{n}\sum_{i=1}^{n}[\phi(x_i)]_j(y_i - \mathbb{E}[Y_i]) \tag{44}$$

By Assumption, we have that $\sum_{i=1}^{n}[\phi(x_i)]_j^2 \leq n$, therefore the random variable $-\frac{1}{n}\sum_{i=1}^{n}[\phi(x_i)]_j y_i$ is $subgaussian(\sigma^2 = 1/n^2 * (\sum_{i=1}^{n}[\phi(x_i)]_j^2) * (1/4)) = 1/(4n)$.

Hence, we have each co-ordinate of $\nabla \mathcal{L}_{n_{\text{logistic}}}(\theta^*)$ is subgaussian$(\sigma^2 = \frac{1}{4n})$. By union bound and subgaussian tail bounds, we have:

$$\mathbb{P}(\|\nabla \mathcal{L}_{n_{\text{logistic}}}\|_\infty \geq \tau) \precsim p^2 \exp(-\tau^2 n c_0) \tag{45}$$

for some constant $c_0$. To get a high probability bound, we essentially want $p^2 \exp(-\tau^2 n c_0) < 1/p^c$ for some $c > 0$. We get that $n >> \frac{1}{\tau^2}\log p$ suffices.

From Lemma 1, we know that $\alpha = 2, \beta = \frac{1}{s\gamma_n\nu_n^2}, \gamma = \infty$, so, we need to put $\tau = \beta = \frac{1}{s\gamma_n\nu_n^2}$. Hence, we get that $n >> s^2\gamma_n^2\nu_n^4 \log p$ samples are enough to ensure that $\|\nabla\mathcal{L}_n(\theta^*)\|_\infty <= \min\left\{\frac{\gamma}{2\kappa}, \left(\frac{1}{2\kappa}\right)^{\frac{\alpha}{\alpha-1}}\beta^{\frac{1}{\alpha-1}}\right\}$.

Now, putting $\tau = c.\sqrt{\frac{\log p}{n}}$ in (45) we get that $\|\nabla\mathcal{L}_{n_{\text{logistic}}}(\theta^*)\|_\infty \precsim \sqrt{\frac{\log p}{n}}$ with high probability, whenever $n \succsim \log p$ Since, we put $\lambda_n = c\|\nabla\mathcal{L}_{n_{\text{logistic}}}(\theta^*)\|_\infty$, we get that $\|\nabla\mathcal{L}_{n_{\text{logistic}}}(\theta^*)\|_\infty + \lambda_n \precsim \|\nabla\mathcal{L}_{n_{\text{logistic}}}(\theta^*)\|_\infty \precsim \sqrt{\frac{\log p}{n}}$. $\qquad\square$

# E    Proof of Lemma 2

*Proof.* From Equation 4, we see that

$$R(\Delta) = (\theta^* + \Delta) - \theta^* - \Delta = 0$$

So, we can choose $\beta = \gamma = \infty$. $\qquad\square$

# F    Proof of Corollary 4

*Proof.* From Lemma 2, we know that $\beta = \gamma = \infty$. Hence, there is no initial sample complexity. We know that $\nabla\mathcal{L}_n(\theta^*) = \theta^* - \frac{1}{n}\sum_{i=1}^{n}x_i$. Each co-ordinate of $[\nabla\mathcal{L}_n(\theta^*)]_j \sim \mathcal{N}(0, 1/n)$. By known concentration results on maximum of $p$ gaussians, we know that when $n >> \log p$, then $\|\nabla\mathcal{L}_n(\theta^*)\|_\infty \precsim \sqrt{\frac{\log p}{n}}$ with high probability. Hence, for individual parameters $\widehat{\theta}_1$ and $\widehat{\theta}_0$ we have $\ell_\infty$ error of $O(\sqrt{\frac{\log p}{n}})$, hence, differential estimate also has $\ell_\infty$-error of $O(\sqrt{\frac{\log p}{n}})$. $\qquad\square$

# G    Proof of Lemma 3

*Proof.* The proof is based on Lemma 5 by [11] and follows from matrix expansion techniques. We provide it's key steps for the sake of completeness. Let $\Delta \in \mathcal{B}_\infty(\gamma) \cap \mathcal{M}(S)$. For $\mathcal{L}_{n_{\text{GGM}}}$, the remainder can be written as:

$$R(\Delta) = (\Theta^* + \Delta)^{-1} - \Theta^{*-1} + \Theta^{*-1}\Delta\Theta^{*-1} \tag{46}$$

$$= (I + \Theta^{*-1}\Delta)^{-1}(\Theta^*)^{-1} - \Theta^{*-1} + \Theta^{*-1}\Delta\Theta^{*-1} \tag{47}$$

$$= \sum_{k=0}^{\infty}(-1)^k((\Theta^*)^{-1}\Delta)^k(\Theta^*)^{-1} - \Theta^{*-1} + \Theta^{*-1}\Delta\Theta^{*-1} \tag{48}$$

$$= \Theta^{*-1}\Delta\Theta^{*-1}\Delta J\Theta^{*-1} \tag{49}$$

where $J = \sum_{k=0}^{\infty} (-1)^k ((\Theta^*)^{-1}\Delta)^k$.

$$\|R(\Delta)\|_\infty = \max_{e_i, e_j} |e_i^T \Theta^{*-1} \Delta \Theta^{*-1} \Delta J \Theta^{*-1} e_j|$$

$$\leq \max_i \left\| e_i^T \Theta^{*-1} \Delta \Theta^{*-1} \right\|_\infty \max_j \left\| \Theta^{*-1} \Delta J \Theta^{*-1} e_j \right\|_1$$

$$\leq \|\Delta\|_\infty \left\| (\Theta^*)^{-1} \right\|_\infty^3 \|\Delta\|_\infty \left\| J^T \right\|_\infty$$

where $J = \sum_{k=0}^{\infty} (-1)^k ((\Theta^*)^{-1}\Delta)^k$.

$$\left\| J^T \right\|_\infty \leq \sum_{k=0}^{\infty} \left\| \Delta(\Theta^*)^{-1} \right\|_\infty^k \leq \frac{1}{1 - \left\| \Delta(\Theta^*)^{-1} \right\|_\infty} \leq \frac{1}{1 - \|\Delta\|_\infty \left\| (\Theta^*)^{-1} \right\|_\infty}$$

Now, by assumption we know that $\Delta \in \mathcal{M}(S)$, hence the support pattern of $\Delta$ is the same as that of $\Theta^*$, which implies that $\|\Delta\|_\infty \leq d \|\Delta\|_\infty$.

So, we have that

$$R(\Delta) = \left\| (\Theta^*)^{-1} \right\|_\infty^3 d \|\Delta\|_\infty^2 \frac{1}{1 - d \|\Delta\|_\infty \left\| (\Theta^*)^{-1} \right\|_\infty}$$

For $\|\Delta\|_\infty < \frac{1}{3d\kappa_{\Sigma^*}}$, we have that $\frac{1}{1 - d\|\Delta\|_\infty \|(\Theta^*)^{-1}\|_\infty} < 1.5$. Hence, we have that

$$R(\Delta) \leq 1.5 d \kappa_{\Sigma^*}^3 \|\Delta\|_\infty^2 \quad \forall \|\Delta\|_\infty < \frac{1}{3d\kappa_{\Sigma^*}}$$

So, the loss is $(2, \frac{1}{1.5 d \kappa_{\Sigma^*}^3}, \frac{1}{3 d \kappa_{\Sigma^*}})$ separable. $\qquad \square$

## H   Proof of Corollary 5

*Proof.* In Lemma 3, we've already established the separability of $\mathcal{L}_{n_{GGM}}$. Assume $\lambda_n = c.\|\nabla\mathcal{L}_n(\theta^*)\|_\infty$. We need to control $\|\nabla\mathcal{L}_{n_{\text{GGM}}}(\theta^*)\|_\infty$, so that $\|\nabla\mathcal{L}_n(\theta^*)\|_\infty <= \min\left\{ \frac{\gamma}{2\kappa}, \left(\frac{1}{2\kappa}\right)^{\frac{\alpha}{\alpha-1}} \beta^{\frac{1}{\alpha-1}} \right\}$. $\qquad \square$

We know that $\nabla\mathcal{L}_{n_{GGM}}(\theta^*) = \widehat{\Sigma}^n - \Sigma^*$, where $\widehat{\Sigma}^n$ is the empirical covariance matrix. To control $\|\nabla\mathcal{L}_{n_{GGM}}(\theta^*)\|_\infty$, we use the following result by [11].

**Lemma 9** (Lemma 1,[11]). *Consider a zero-mean random vector $(X_1, \ldots, X_p)$ with covariance such that each $X_i/\sqrt{\Sigma_{ii}^*}$ subGaussian with parameter $\sigma^2$. Given $n$ iid samples, then the associated sample covariance $\widehat{\Sigma}_n$ satisfies the tail bound(modulo constants)*

$$\mathbb{P}\left( |\widehat{\Sigma}_{ij}^n - \Sigma_{ij}^*| > \delta \right) \leq 4 \left( -cn\delta^2 \right)$$

Taking union bound over $p^2$ terms, we get that, for $n >> \{\max\left\{ \frac{4\kappa^2}{\gamma^2}, (2\kappa)^{2\alpha/(\alpha-1)}(\frac{1}{\beta})^{2/(\alpha-1)} \right\} \log p$, with high probability,

$$\|\nabla\mathcal{L}_{n_{GGM}}(\theta^*)\|_\infty = \left\| \widehat{\Sigma}_{ij}^n - \Sigma_{ij}^* \right\|_\infty \leq \min\left\{ \frac{\gamma}{2\kappa}, \left(\frac{1}{2\kappa}\right)^{\frac{\alpha}{\alpha-1}} \beta^{\frac{1}{\alpha-1}} \right\}$$

From Lemma 3, we know that $\alpha = 2, \beta = \frac{1}{1.5\kappa_{\Sigma^*}d}, \gamma = \frac{1}{2d\kappa_{\Sigma^*}}$. Plugging these values, we get that for $n >> d^2 \log p$, $\left\| \widehat{\Theta}_1 - \Theta_1^* \right\|_\infty \precsim \sqrt{\frac{\log p}{n}}$ and similarly, $\left\| \widehat{\Theta}_0 - \Theta_0^* \right\|_\infty \precsim \sqrt{\frac{\log p}{n}}$.

Moreover from Theorem 1, we get that $support(\widehat{\Theta}_i) \subseteq support(\Theta_i^*)$ for $i = 0, 1$.

By triangle inequality, $\left\| \widehat{\Theta}_{\text{diff}} - \Theta_{\text{diff}}^* \right\|_\infty \precsim \sqrt{\frac{\log p}{n}}$.

# I Proof of Lemma 4

*Proof.* This statement is a special instantiation of Theorem 1 in [7]. A proof of the lemma can also be found in [4](Theorem 7). □

# J Cross-Validated Experiments

(a) $s = 4, p = 512$    (b) $s = 16, p = 512$    (c) $s = 64, p = 512$

Figure 3: Effect of sparsity on Classification Error. Observe the effect of $s$ on the burn-in period for logistic before it starts classifiying;. All experiments plotted the average of 20 trials. In all experiments we set the regularization parameter $\lambda_n$ using cross-validation.