[Reviews · NeurIPS 2017]

Reviewer 1



This paper compares the convergence rates of M estimators for differential parameter estimators. Two classes of models are considered: generative and discriminative ones. Generative models estimate (canonical) parameters of two distinct sets of samples, and compute the difference. Discriminative models directly estimate the difference based on Bayes rule and logistic regression. Convergence rates are established by introducing a new concept of local separability: a measure of how much, well, and wide a loss function behaves as separable. Results cover low and high dimensional settings, the latter being treated with l1-sparsity regularization. From my understanding, the generative method is in general better in both theory and simulations. The paper is well written, well organized and theory look sound. Nevertheless, as I am not expert in this field, I cannot judge about the originality and implications of these results. I just have the following remarks: - line 75: x_i^{(0} -> x_i^{(0)} - line 94, eq (5): why not writing mu instead of theta? To be precise it is the negative average log-likelihood (up to constants). Same for eq (6). - line 95, eq (6): trace(Theta, hatSigma) -> trace(Theta hatSigma) (no comma) - line 98 to 104: should theta_2^* be theta_0^*? - line 102, eq (8): why (8) does not depend on c^*? - line 103: phi(t) should be Phi(t) or in bold. - line 158: Missing a period after "Irrepresentability". - line 208, Lemma 3: I could not find the definition of d^*_Theta and kappa_Sigma^*. - line 250: Missing a dash between "soft" and "Thresholding". - line 259: Missing a period before "For any classifier". - line 289: Missing a period before "In this setting".

Reviewer 2



This paper starts by developing a notion of local "separability" of a loss function, which they use to get l_infty convertence rates, in terms of the separability parameters, for low and high dimensional settings. These rates are then applied to then applied to a probabilistic classification problem with both a generative and discriminative approach. After computing the teh separability parameters for each, they can apply the theorems to get l_infty convergence rates for the discriminative approach (logistic regression), as well as two generative approaches (for the cases that x|y is isotropic Gaussian and gaussian graphical model). They next consider l_2 convergence rates. The discriminative rate is trivial based on the support consistency and the l_infty rates. The naive generative algorithm would not have good scaling with the number of dimensions, but with a soft-thresholding operator, they can get support consistency and good rates. Finally they show the effect of sparsity on sample complexity for the discriminative and the thresholded generative approaches. As predicted by their theory, logistic rate suffers compared to the generative approach, especially with less sparsity. This seems to be impressive work. I have a few questions / requests for clarifications and notes: Notes - Terminology temporarily switches from empirical loss function to empirical risk function on line 127 on page 4 - I'm a little confused by the comment on lines 181-182 page 5: "Theorems 1 and 2 give us an estimate on the number of samples required by a loss to establish convergence. In general, if gamma is very small or beta is very small then we would need a lot of samples." Aren't beta and gamma dependent on L_n, and thus the number of samples? Is this really about the behavior of beta and gamma as a function of n? - Probably the same confusion again: In several places (e.g. line 194, 195 on page 6, cor 3), we have inequalities with dependence on n on both sides of the equations. We're trying to get convergence rates in terms of n, but won't the separability parameters have their own behavior w.r.t. n. Where is that captured? - It's claimed that Theorems 1 and 2 give us l_infty convergence rates. Embarrassingly, I don't see how we get the rates. Do we need to know how the separability changes with n? - “From Cor 3 and 4, we see that for IG, both the discriminativce and generative approach achieve the same convergence rate, but the generative approach does so with only a logarithmic dependence on the dimension.” In what way is logistic different? Also corollary 4 has a n >> log p -- is that right? Corollary 3 has the notation \succsim -- I'm not familiar with that notation, and perhaps should be defined in a footnote?

Reviewer 3



The paper compares generative and discriminative models for general exponential families, also in the high-dimensional setting. It develops a notion of separability of general loss functions, and provides a framework for obtaining l_{infty} convergence rates for M-estimators. This machinery is used to analyze convergence rates (both in l_{infty} and l_2) of generative and discriminative models. I found the paper quite hard to follow, but this might potentially be also due to my lack of expertise in high dimensional estimation. While the introduction, as well as background and setup (Section 2) are accessible and quite easy to follow, in the following sections the paper becomes quite dense with equations and technical results, sometimes given without much guidance to the reader. Sometimes authors' comments made me quite confused; for instance, in lines 215-218, Corollary 4 and 3 are compared (for discriminative and generative models) and it is stated that "both the discriminative and generative approach achieve the same convergence rate, but the generative approach does so with only a logarithmic dependence on the dimension", whereas from what I can see, the dependence on the dimension in both Corollary 3 and 4 is the same (sqrt(log p))? The core idea in the paper is the notion of separability of the loss. The motivating example in the introduction is that in the generative model with conditional independence assumption, the corresponding log-likelihood loss is fully separable into multiple components, each of which can be estimated separately. In case of the corresponding discriminative model (e.g. logistic regression), the loss is much less separable. This motivates the need to introduce a more flexible notion of separability in Section 3. It turns out that more separable losses require fewer samples to converge. Unfortunately, I am unable see this motivation behind the definition of separability (Definition 1 and 2), which states that the error in the first order approximation of the gradient is upper bounded by the error in the parameters. I think it would improve the readability of the paper if the authors could motivate why is this particular definition used here and how is it related to the intuitive notion of separability of the loss into multiple components. Starting from Corollary 3, the authors use notation "\succsim" without explanation. What does it mean? It is hard for me to judge the significance of the paper, as I not familiar with previous work in this setup (apart from the work by Ng & Jordan), but I found the results quite interesting, if they were only presented more clearly. Nevertheless, I evaluate the theoretical contribution to be quite strong. What I found missing is a clear message about when to use generative vs. discriminative methods in high dimension (similar to the message given by Ng & Jordan in their paper). Small remarks: 36: "could be shown to separable" -> "could be shown to be separable" 65: n -> in 87: : and j should be in the subscript 259: bayes -> Bayes 254-265: the "Notation and Assumptions" paragraph -- the Bayes classifier is defined only after its error was already mentioned.

Reviewer 4



The paper presents a generalization to previous work on ellinfty recovery in high-dimensional models. Past results exist but only for the setting of linear regression. The authors generalize the existing results to that of general loss functions. I did not have enough time to thoroughly go through the proofs, but they seem correct. I would say that parts of the paper are haphazardly written, and so the other should focus a lot of effort on fixing the presentation. Comments: More discussion of the proposed links to incoherence. Incoherence is of course a much stronger assumption used in the linear setting to establish ellinfty bounds. Can analogous notions of incoherence be developed in this setting? The authors should look at the work of Ren et. al. (ASYMPTOTIC NORMALITY AND OPTIMALITIES IN ESTIMATION OF LARGE GAUSSIAN GRAPHICAL MODELS) to contrast against ellinfty bounds and optimal rates as well as assumptions for precision matrix recovery. A lot of the text following Corollary 3 and beyond isn't clear. For instance, under isotropic Gaussian design, ellinfty control for classification under mild assumptions on the correlation between the labels and the design allows one to identify the support with only n > log p samples. What is the connection to generative versus discriminative models here? Some remarks that suggest one method requires more samples than another (e.g. line 218) should be softened since there aren’t lower bounds.

Reviewer 5



In this paper, the convergence of maximum likelihood estimates w.r.t. sample complexities, for generative as well as discriminative models, is studied. I find this paper interesting. It is well written, though some scope for improvements in the terms of exposition. The authors find that generative models can outperform discriminative methods, under certain sparsity related conditions, if the number of samples is small while the dimension of data is high. The theoretical analysis is claimed to be novel, and should have an impact for real world problems in the long run. Logistic regression is considered as an example discriminative model. For generative models, Gaussian and Gaussian Graphical models are considered. The authors managed to keep the exposition such that it is of interest to a general machine learning audience, which can be somewhat hard for a theoretical paper. I have a few suggestions. (1) Is it possible to draw a figure explaining the idea of separability, in reference to the Definition 1 ? (2) I didn't understand the argument written before the Corollary 5, w.r.t. the comparison between Corollary 3 and 4. (3) There are some notation mistakes. On a separate note, it would be nice to extend the paper, with more simulations, and put the extended version on ArXiv, etc.